# Post-Quantum and Code-Based Cryptography—Some Prospective Research Directions

**Chithralekha Balamurugan [1], Kalpana Singh [2,\*], Ganeshvani Ganesan [1] and Muttukrishnan Rajarajan [3]**

[1] Department of Computer Science, Pondicherry University, Chinna Kalapet, Kalapet,
Puducherry 605014, India; tchithralekha.csc@pondiuni.edu.in (C.B.); vaniganesh0306@gmail.com (G.G.)

[2] IRT SystemX, 2 Boulevard Thomas Gobert, 91120 Palaiseau, France

[3] Institute for Cyber Security, University of London, Northampton Square, Clerkenwell,
London EC1V 0HB, UK; r.muttukrishnan@city.ac.uk

[\*] Correspondence: kalpana.singh@irt-systemx.fr; Tel.: +33-755962727

**Abstract:** Cryptography has been used from time immemorial for preserving the confidentiality of data/information in storage or transit. Thus, cryptography research has also been evolving from the classical Caesar cipher to the modern cryptosystems, based on modular arithmetic to the contemporary cryptosystems based on quantum computing. The emergence of quantum computing poses a major threat to the modern cryptosystems based on modular arithmetic, whereby even the computationally hard problems which constitute the strength of the modular arithmetic ciphers could be solved in polynomial time. This threat triggered post-quantum cryptography research to design and develop post-quantum algorithms that can withstand quantum computing attacks. This paper provides an overview of the various research directions that have been explored in post-quantum cryptography and, specifically, the various code-based cryptography research dimensions that have been explored. Some potential research directions that are yet to be explored in code-based cryptography research from the perspective of codes is a key contribution of this paper.

**Keywords:** quantum computing; post-quantum cryptography; code-based cryptography; cryptosystem; cryptography; privacy





## 1. Introduction

Cryptographic systems are built on complex mathematical problems such as integer factorization and computing discrete logarithms [1,2], which can only be solved if knowledge of some secret data is available; typically a very large number. Without these numbers, it is impossible to reverse-engineer encrypted data or create a fraudulent digital signature. These numbers are what we know as cryptographic keys. For instance, the RSA algorithm [3] works by using pairs of very large prime numbers to generate public and private keys. The public key can be used to create a mathematical challenge which can only be solved by someone who holds the private key. Attempting to guess the answer, by way of a brute-force search, would take thousands of years using contemporary computers. Unlike their classical counterparts, quantum computers will be able to solve these mathematical problems incredibly quickly. The asymmetric algorithms we use today for digital signatures and key exchange will no longer be strong enough to keep data secret once a sufficiently powerful quantum computer can be built.

This means that core cryptographic technologies that we have to rely on, RSA and elliptic curve cryptography, will become insecure. Even symmetric cryptographic algorithms have now been found possible to be attacked by quantum computers [4]. This context alludes to the fact that both symmetric and asymmetric algorithms which are in widespread use today can succumb to quantum attacks and hence, quantum attack resistant, or in other words, post-quantum cryptographic algorithms, need to be evolved.

Certain candidate families of post-quantum schemes have been realized including code-based [5], hash-based [6], multivariate [7], lattice-based [8,9] and isogeny-based [10] solutions. The maturity in post-quantum research has led to the formulation of various post-quantum cryptographic algorithms, the standardization of post-quantum algorithms by various standardization bodies world-wide, industry adoption of post-quantum technology, and the development of open-source post-quantum libraries.

In this paper, an overview of these research dimensions that have been explored in post-quantum cryptography has been provided. Furthermore, with a specific focus on code-based cryptography (CBC), the key milestones in CBC research have been provided. Although code-based cryptography focuses on the use of codes, other potential research dimensions could be explored in CBC research leveraging the codes—which is yet to receive focus in CBC research. Delineating this aspect has been the important contribution of this paper, and has been achieved by the study of the linear codes, their operations and relationships, and other kinds of codes that could be deployed in CBC. This paper also highlights that privacy-preserving CBC is another prospective research dimension that is yet to receive enough focus in CBC research.

Section 2 briefs about quantum computing and alludes to the motivation for post-quantum cryptographic research. Section 3 reviews post-quantum research dimensions with due summaries and comparisons. Section 4 explains the CBC research dimensions that have been explored. Section 5 describes the white spaces that are yet to be addressed in CBC research from the perspective of codes and privacy aspects. Section 6 concludes the paper.

## 2. Quantum Computing

A quantum computer is a machine which employs quantum-physical phenomena to perform computations in a way that is fundamentally different from a "normal", classical computer [11]. Whereas a classical computer is, at any point in time, in a fixed state—such as a bit string representing its memory contents—the state of a quantum computer can be a "mixture", a so-called superposition, of several states. Classical computers carry out logical operations using the definite position of a physical state. These are usually binary, meaning its operations are based on one of two positions. A single state—such as on or off, up or down, 1 or 0—is called a bit. Note that the internal state is hidden: the only way to obtain information about the state is to perform a measurement, which will return a single non-superimposed classical output, such as a bit string, that is randomly distributed according to the internal state, and the internal state is replaced by the measurement outcome.

The work was initiated by several mathematicians and physicists such as Paul Benioff (1980) [12], Yuri Manin (1980) [13], Richard Feynman (1982) [14], and David Deutsch (1985) [15]. With decades of research, the development of quantum computing has been challenging yet groundbreaking.

Thus, quantum computing constitutes a new computing paradigm, which is expected to solve complex problems that require far more computational power than what is possible with the current generation of computer technologies. Advance research in materials science, molecular modelling, and deep learning are a few examples of complex problems that quantum computing can solve.

*Shor's 1994 and Grover's 1996 Algorithms*

On the quantum algorithmic development, there are two groundbreaking algorithms which have laid out a strong foundation towards breaking today's number of theoretically based public-key cryptosystems. In 1994, Shor proposed a polynomial-time (efficient) algorithm [16] for solving integer factorization and discrete logarithm problems. The algorithm relies on the existence of quantum computers, and hence this type of algorithm is called quantum algorithms in this article. Shor's quantum algorithm and its variants can be used for breaking most of the currently used public-key cryptosystems.

In 1996, Grover proposed an $O(\sqrt{N})$-query complexity of quantum algorithm for functions with $N$-bit domains [17]. This quantum algorithm once realized on quantum

computers can be used for breaking symmetric-key cryptosystems, and to defend against attacks based on Grover's algorithm, we need to double the key sizes in order to achieve a similar level of security against conventional computers.

For example, for 128-bit symmetric-key security, we need to use symmetric-key cryptosystems which are originally designed for achieving 256-bit security against attacks based on Grover's quantum algorithm. It is also predicted that quantum computers will be able to break several of today's cryptographic algorithms that are used to secure communications over the Internet, provide the root of trust for secure transactions in the digital economy and encrypt data. To protect against attacks from quantum computers, vendors of security products and service providers must constantly assess the risk associated with the choice of cryptographic algorithms and evolve entirely new quantum-resistant algorithms for the post-quantum world.

## 3. Post-Quantum Cryptography

Post-quantum cryptography (PQC) is about devising cryptographic algorithms that are secure in the quantum era with security against both classical/conventional and quantum computers. There are several candidate approaches for building the post-quantum cryptographic schemes, as described below in Section 3.1 [18–20].

### 3.1. Post-Quantum Cryptography Candidates

This subsection delineates the candidates of the PQC schemes. These are hash-based cryptography, code-based cryptography, multivariate cryptography, lattice-based cryptography, and isogeny-based cryptography schemes.

### 3.1.1. Hash-Based Cryptography

Hash-based cryptography focuses on designing digital signature schemes based on the security of cryptographic hash functions, e.g., SHA-3. These schemes are based on the security of hash functions (as a one-way function, collision-resistant property, and hardness of second pre-image attacks), and require fewer security assumptions than the number-theoretic signature schemes (e.g. RSA, DSA). Ralph Merkle in 1989 introduced Merkle Signature Scheme (MSS) [21], which is based on one-time signatures (e.g., the Lamport signature scheme) and uses a binary hash tree (Merkle tree). The MSS is resistant to quantum computer algorithms. More details can be found in this survey on hash-based schemes Butin (2017) [22]. Sphincs+ hash-based signature [23] is chosen as an alternate solution in the outcome of the third round of the NIST standardization process [24].

### 3.1.2. Code-Based Cryptography

Code-based cryptography [25,26] has its security relying on the hardness of problems from coding theory, for example, syndrome decoding (SD) and learning parity with noise (LPN). These cryptosystems are based on error-correcting codes to construct a one-way function. The security is based on the hardness of decoding a message which contains random errors and recovering the code structure. A classic McEliece code-based encryption scheme [5] is chosen as a finalist scheme in the outcome of the third round of the NIST standardization process [24].

### 3.1.3. Multivariate Cryptography

Multivariate cryptography has its security relying on the hardness of solving multivariate systems of equations. These schemes are based on systems of multivariate polynomial equations over a finite field $F$. There are several variants of multivariate cryptography schemes based on hidden field equations (HFE) trapdoor functions, such as the unbalanced oil and vinegar cryptosystems (UOV). UOV is used for signatures. Other examples of multivariate cryptography are Rainbow, TTS, or MPKC schemes. More about the current state of the multivariate cryptography schemes can be found in the paper by Ding and Petzoldt (2017) [27]. Two multivariate signature schemes are chosen in the outcome of

the third round of the NIST. Rainbow [28] is one of the finalists. GeMSS [29] is one of the alternate finalist schemes.

### 3.1.4. Lattice-Based Cryptography

Lattice-based cryptography seems to be one of the most active directions in recent years, for several key reasons. First, it has strong security guarantees from some well-known lattice problems, for example, shortest vector problem (SVP) and the ring learning with errors (RLWE) problem [30]. Second, it enables powerful cryptographic primitives; for example, fully homomorphic encryption (FHE) and functional encryption [31]. Third, some new lattice-based cryptographic schemes have become quite practical recently, for example, the key exchange protocol NewHope [32], and a signature scheme BLISS [33].

Lattice-based cryptography is one of the successful schemes in the third round result of the NIST standardisation process. Kyber [34], NTRU [35], SABER [36] lattice-based encryption schemes are chosen as the finalists' schemes. NTRUprime [37] is in the alternate finalist lattice-based encryption scheme. Dilithium [38] and Falcon [39] lattice-based signature schemes are also finalist schemes.

### 3.1.5. Isogeny-Based Cryptography

Isogeny-based cryptography is a specific type of post-quantum cryptography that uses certain well-behaved maps between abelian varieties over finite fields (typically elliptic curves) as its core building block. Its main advantages are relatively small keys and its rich mathematical structure, which pose some extremely interesting questions to cryptographers and computer allegorists. These schemes are based on supersingular elliptic curve isogenies [10] that are secure against quantum adversaries. These schemes are secured under the problem of constructing an isogeny between two supersingular curves with the same number of points. Isogeny-based schemes may serve as digital signatures or key exchange, such as the supersingular isogeny Diffie–Hellman (SIDH) scheme [40]. SIKE [41] is the only isogeny-based encryption scheme in the alternate list of NIST third round results. There is no isogeny-based signature scheme identified in the NIST third round outcome.

### 3.1.6. Comparison of Post-Quantum Cryptography Algorithms

The five post-quantum cryptographic algorithm categories have been compared considering only the key algorithm(s) in each category, and the comparison has been presented in Table 1. This is to provide a quick summary of the key algorithms in the five post-quantum algorithm categories, and is not intended for a complete review of the existing algorithms in each of the post-quantum algorithm categories. An earlier comparison of post-quantum cryptographic algorithms has also been attempted in [42].

### 3.2. Industry Adoption of Post-Quantum Cryptography

The industry adoption of post-quantum cryptography is happening very aggressively. On this front, the following lines of works are found to be available:

- Industry survey of post-quantum cryptography,
- Revenue Assessment of post-quantum cryptography,
- Industry initiatives in PQC—PQC R&D, PQC-based products, PQC products, PQC consulting.

**Table 1.** Comparison of PQC algorithms.

| Serial Number | Post-Quantum Algorithm Category | Post-Quantum Algorithms available in this Category | Name of Most Prevalent Algorithm | Type of Algorithm - Encryption/ Signature/ Key Exchange | Public Key Size | Private Key Size | Signature Size | Strengths | Weaknesses | included in open-source library Liboqs | Attacks | Other Detailed Comparisons |
|---|---|---|---|---|---|---|---|---|---|---|---|---|
| 1 | Lattice Based Cryptogaphy | 1. Encryption/ Decryption 2. Signature 3. Key Exchange (RLWE) | NTRU Encrypt | E | 6130 B | 6743 B | – | 1. More efficient encryption and decryption, in both hardware and software implementations 2. Much faster key generation allowing the use of disposable keys . 3. low memory use allows it to use in applications such as mobile devices and smart-cards. | 1. Complexity is high in NTRU 2. There is the possibility of the occurrence of a decryption failure from a validly created cipher-text | ✓ | Brute Force attack, meet-in-middle attack, lattice reduction attack, chosen cipher text attack | Gaithru et al. 2014 [43] |
| | | | BLISS II (Bimodal Lattice Signature Scheme) | S | 7 KB | 2 KB | 5 KB | | | ✗ | Side Channel Attack, Branch tracing attack, Rejection Sampling, Scalar Product Leakage | Espitau et al. 2017 [44] |
| 2 | Multi-Variate | Signature only | Rainbow | S | 124 KB | 95 KB | 424 KB | It is based on the difficulty of solving systems of multivariate equations | Only Signature Scheme is available | ✓ | Direct Attack, Min Rank Attack, High Rank Attack, UOV Attack, UOV Reconciliation attack, Attacks against hash function | Petzoldt et al. 2010 [45] |
| 3 | Hash Based Signature | Signature Only | SPHINCS | S | 1 KB | 1 KB | 41 KB | 1. Best alternative to number theoretic signature 2. Small and medium size signatures 3. Small Key size | Only Signature Scheme is available Speed | ✗ | Subset Resilience, One-wayness, Second Pre-image resistance, PRG, PRF and undetectability, Fault Injection Attacks | Bernstein et al. 2015 [46] |
| | | | SPHINCS+ | S | 32 B | 64 B | 8 KB | | | ✓ | Distinct-function multi-target second-preimage resistance, Pseudorandomness (of function families), and interleaved target subset resilience, timing attack, differential and fault attacks | Bernstein et al. 2019 [47] |
| 4 | Super-singular elliptic curve isogeny cryptography | Key Exchange Only | Supersingular Isogeny Diffie Helman (SIDH) | K | 751 B 564 (compressed SIDH) | 48 B 48 (compressed SIDH) | – | Difficulty of computing isogenies between supersingular elliptic curves which is immune to quantum attacks | Cannot be used for non-interactive key exchange, can only be safely used with CCA2 protection | ✓ | Side-channel attacks, Auxiliary points active attack, adaptive attack | Costello et al. 2016 [41] |
| 5 | Code-Based Cryptography | 1. Encryption/Decryption 2. Signature | Classic McEliece Cryptosystem | E | 1 MB | 11.5 KB | – | One of the cryptosystem which is successful till the third round of NIST Post-Quantum algorithm standardization process | Very Large Key size | ✓ | Structural Attack, Key recovery attack, Squaring Attack, Power Analysis Attack, Side Channel attack, Reaction attack, Distinguishing attack, message recovery attack | Tillich 2018 [48], Repka et al. 2014 [49] |

### 3.2.1. Industry Survey

The industry survey has been carried out by Digicert involving IT Directors, IT Generalists, IT security professionals, and other professionals belonging to the USA, Germany, and Japan. The survey focused on identifying the following:

- The awareness or the understanding about PQC with industry professionals,
- The industry professionals' prediction of timelines by which quantum computers would break the existing modular arithmetic cryptographic algorithms,
- The understanding among the industry professionals about the significance of threat imposed by quantum computing on existing cryptographic algorithms,

- The study of industry readiness to adopt PQC.

The results obtained indicate that the awareness of PQC among industry professionals is reasonably good and they have a clear understanding of an appropriate timeline by which quantum computers would break the existing cryptographic algorithms. The impact of the threat imposed by PQC is also well perceived by the industry professionals. The survey also reveals the industry readiness in the adoption of PQC to be beyond 50%. These aspects indicate that the industry survey has helped to comprehend the line of thought and the industry awareness and preparedness for PQC among industry professionals; this is very important when it comes to starting to be precautious and working out plans for adoption of PQC given the discernment of the threat due to quantum computing technology.

### 3.2.2. Revenue Assessment of Post-Quantum Cryptography

As per the ten-year Market and Technology Forecast Report in [50], a comprehensive study about the prospective markets for PQC products and services has been carried out. The IT industry, cybersecurity industry, telecommunications industry, financial services industry, healthcare industry, manufacturing industry, PQC in IoT, and public sector applications of PQC have been identified as prospective markets for PQC in the report. An elaborate study of how PQC could augment or enhance the functioning of the above industries has been detailed in the report. A ten-year forecast of revenue assessment of PQC in each of the above industries is detailed in the said report, which is indicative of the prospective industry market and trend for PQC.

### 3.2.3. Industry Initiatives in PQC–PQC R& D, PQC-Based Products, PQC Products, PQC Consulting

The IT industry has been closely following up with post-quantum cryptographic research and the standardization process. The industry initiatives could be observed in terms of the following.

- Research and development in post-quantum cryptography—organizations like IBM [51], Microsoft [52], etc. conduct research and development in post-quantum cryptography.
- Development of post-quantum-based products—for example, Avaya has tied up with post-quantum (a leading organization developing post-quantum solutions), to incorporate post-quantum security into its products.
- PQC products—Organizations like Infineon, Qualcomm (OnBoard Security), Thales, Envieta, etc. have developed post-quantum security hardware/software products [50].
- Post-quantum consulting—Utimaco is one of the leading players which provides for post-quantum cryptography consulting.

Table 2 provides a comparison of the various industry initiatives of PQC concerning PQC-, R&D-, PQC-based security products development, PQC product development, and PQC consulting.

**Table 2.** Summary of Industry Initiatives in Post-Quantum Cryptography.

| Serial Number | Name of the Organization | Country | Type of PQC Work Involved | Algorithms Used | Collaborator | PQC Product Developed |
|---|---|---|---|---|---|---|
| 1. | Avaya [53] | USA | PQC based products | | Post-Quantum | Quantum-safe messaging, voice calls and document sharing |
| 2. | Envieta Systems [54] | USA | PQC products, PQC Consulting | | – | Developed Hardware and Software Post-Quantum Implementation Cores including those for embedded systems as well |
| 3. | Google [55] | USA | PQC products | HRSS-SXY (variant of NTRU encryption) and SIKE (supersingular isogeny key exchange) | Cloudfare | Post-quantum cryptography˙encryption and signature methods for chrome browser |
| 4. | IBM [51] | Switzerland | R&D | Lattice-based Cryptography | – | Quantum safe Cloud and Systems |
| 5. | Infineon [56] | Germany | PQC products | Variant of New Hope Algorithm | – | Implemented a post-quantum key exchange scheme on a commercially available contactless smart card chip Post-Quantum security for Government Identity Documents,ICT technology, Automotive Security,Communication Protocols |
| 6. | Isara [56] | Canada | PQC products | Hierarchical Signature Scheme (HSS) and eXtended Merkle Signature Scheme (XMSS) | Futurex, Post-Quantum | ISARA Radiate, Quantum-safe Toolkit is a high-performance, lightweight, standards-based quantum-safe software development kit, built for developers who want to test and integrate next-generation post-quantum cryptography into their commercial products |
| 7. | Microsoft Research [52] | USA | PQC products | FrodoKEM , SIKE, Picnic, QTesla | – | Post-Quantum SSH, TLS, VPN |
| 8. | Qualcomm/ OnBoard Security [57] | USA | PQC based products | pqNTRUsign | OnBoard Security | OnBoard Security has developed a digital signature algorithm that can resist all known quantum computing attacks. pqNTRUsign will replace RSA and ECDSA, the most commonly used quantum-vulnerable signature schemes. |

### 3.3. Standardization Efforts in PQC

The standardization of post-quantum algorithms has been taken up by different standardization bodies across the globe. The following section provides an overview of the standardization activities taken up by the following standardization bodies viz. NIST, ITU, ISO, ETSI, and CRYPTREC.

### 3.3.1. NIST

NIST is one of the primary bodies involved in the standardization of post-quantum algorithms [58]. The standardization process began in 2016, and it is currently in the third round after two previous rounds of post-quantum algorithm evaluations. The third round finalists [24] comprise 4 public-key encryption algorithms and 3 digital signature algorithms, along with 5 and 4 alternate candidates for public-key encryption and digital signature algorithms, respectively. These are listed in the Table 3. For standardization, the algorithm submissions were first ensured to fulfill certain minimum acceptability requirements before evaluation, and evicted otherwise. In each round, the evaluations were carried out using a set of criteria under security, cost, and efficiency concerning algorithm implementation aspects, respectively. From Table 3, it is obvious that lattice-based post-quantum technology has the majority contribution among the list of standardized post-quantum algorithms.

**Table 3.** NIST Standardization Efforts [59].

| Post-Quantum Algorithm Type | Third Round Finalist | Technology | Alternate Candidates | Technology |
|---|---|---|---|---|
| Public Key Encryption/ Key Encapsulation Mechanisms | Classic McEliece | Code | BIKE | Code |
| | CRYSTALS KYBER | Lattice | FrodoKEM | Lattice |
| | NTRU | Lattice | HQC | Code |
| | SABER | Lattice | SIKE | Supersingular Isogeny |
| Digital Signature Algorithms | CRYSTALS–DILITHIUM | Lattice | GeMSS | Multivariate Polynomial |
| | FALCON | Lattice | PICNIC | Other |
| | RAINBOW | Multivariate Polynomial | SPHIMCS+ | Hash |

### 3.3.2. International Telecommunication Union (ITU)

International Telecommunication Union (ITU) has two study groups SG13 and SG17 for PQC and a focus group FG QIT4N [60], which works on pre-standardization activities. SG13 has provided a plethora of standards under the categories of (i) Architecture, Framework, Function of Quantum Key Distribution Network and (ii) Quality of Service of Quantum Key Distribution Network. Some of these standards are published and many are work-in-progress. SG17 has provided a set of standards related to the security aspects of the Quantum Key Distribution Network. FG QIT4N group focuses on the pre-standardization activities related to Quantum Information Technology for Networks. It has two working groups, WG1 and WG2; working on this and relevant technical reports has been published by these working groups. A detailed listing of the standards and pre-standardization reports of ITU could be found in [60].

### 3.3.3. European Telecommunications Standards Institute (ETSI)

European Telecommunications Standards Institute (ETSI) ETSI develops ETSI group specifications and group reports describing quantum cryptography for ICT networks. The ETSI has been involved in the standardization activities of QKD since 2008. The standards developed by ETSI pertain to Quantum-Safe Cryptography, CYBER, and Quantum Key Distribution. Various standards under the said three categories have been developed by ETSI. The listing and details of the standards under the said categories could be found in [61].

### 3.3.4. ISO

ISO/IEC JTC 1/SC 27 IT Security techniques include 5 working groups [60,62]. SC27 has developed many cryptography standards in the past 28 years. In ISO/IEC JTC 1/SC27, WG2 is for the standardization of cryptography and security mechanisms. The standards cover a large scope, from relatively advanced topics such as homomorphic encryption, group signatures to some essential functions such as block ciphers and hash functions. A six-month study period on Quantum Resistant Cryptography was initiated at a SC 27/WG 2 meeting held in Jaipur, India October 2015. After the first six months, the study period was extended three times and determined to close at SC 27/WG 2 meeting held in Berlin, Germany, in November 2017. As a result of the study period, it was determined to generate a WG2 standing document (SD). An outcome of this post-quantum cryptography study is SD8. SD8 provides a survey on different categories/families of post-quantum cryptography, and is intended to prepare WG2 experts for standardization. SD8 is created in multiple parts, where each part corresponds to each of the post-quantum cryptographic techniques vires. Hash, Lattice, Code, etc. ISO/IEC JTC1 SC27 WG3 (ISO/IEC 23837) focus on security requirements, test, and evaluation methods for quantum key distribution. This addresses QKD implementation security issues. A high-level framework for the

security evaluation of the QKD module under the Common Criteria (CC) (ISO/IEC 15408) framework has evolved.

### 3.3.5. CRYPTREC

CRYPTREC is the Cryptography Research and Evaluation Committees set up by the Japanese Government to evaluate and recommend cryptographic techniques for government and industrial use. The following activities have been carried out as part of CRYPTREC. The CRYPTREC cipher list was published in 2013, followed by the CRYPTREC Report 2014 on lattice problems. The Cryptanalysis Evaluation Working Group was formed in 2015. This group has published a Report on PQC in 2018. CRYPTREC plans to revise the CRYPTREC Cipher List to include post-quantum algorithms by 2022–2024. The details of the above activities are detailed in [63].

Table 4 provides a summary of all the standardization efforts described.

**Table 4.** Summary of standardisation efforts in Post-Quantum Cryptography.

| Parameter | Country | Focus Area of Standardization in PQC | Function Standards/QOS Standards | Status |
|---|---|---|---|---|
| NIST | USA | Quantum resistant Algorithm Standardization for Cryptography, Key encapsulation mechanism, digital signature | Function Standards | Round 1 and Round 2 of standardization process is completed. Round 3 in progress |
| ETSI | Europe | Quantum Safe Cryptography, Quantum Key Distribution | Function Standards | Published standards |
| ISO | NA (A Non-Governmental Organization) | Quantum Key Distribution | Function Standards | Work in progress—Standards yet to be published |
| ITU | A specialized agency of United Nations | Quantum Key Distribution | Function and QOS Standards | Only two standards published. Others are work in progress |
| CRYPTREC | Japan | Quantum resistant Algorithm Standardization for Cryptography, Key encapsulation mechanism, digital signature | Function Standards | List of standardized algorithms are expected to be published between 2022-2024 |

In addition to the above efforts, the following works are being carried out by other organizations for PQC:

- IETF has formulated a Framework to Integrate Post-quantum Key Exchanges into Internet Key Exchange Protocol Version 2 (IKEv2), refs. [64,65].
- libpqcrypto [66] is a new cryptographic software library produced by the PQCRYPTO project. libpqcrypto collects this software into an integrated library, with (i) a unified compilation framework; (ii) an automatic test framework; (iii) automatic selection of the fastest implementation of each system; (iv) a unified C interface following the NaCl/TweetNaCl/SUPERCOP/libsodium API; (v) a unified Python interface (vi) command-line signature/verification/encryption/decryption tools, and (vii) command-line benchmarking tools.
- The Cloud Security Alliance Quantum-Safe Security Working Group's [67] goal is to address key generation and transmission methods that will aid the industry in understanding quantum-safe methods for protecting data through quantum key distribution (QKD) and post-quantum cryptography (PQC). The goal of the working group is to support the quantum-safe cryptography community in the development and deployment of a framework to protect data, whether in movement or at rest. Several reports and whitepapers on quantum safe cryptography have been published.

- NSA is publicly sharing guidance on quantum key distribution (QKD) and quantum cryptography (QC) as it relates to secure National Security Systems (NSS). NSA is responsible for the cybersecurity of NSS, i.e., systems that transmit classified and/or otherwise sensitive data. Due to the nature of these systems, NSS owners require especially robust assurance in their cryptographic solutions; some amount of uncertainty may be acceptable for other system owners, but not for NSS. While it has great theoretical interest, and has been the subject of many widely publicized demonstrations, it suffers from limitations and implementation challenges that make it impractical for use in NSS operational networks.

### 3.4. Post-Quantum Cryptography Tools and Technology

Numerous open-source projects and libraries implement quantum-safe schemes with free public access to the source code and packages. This subsection provides a walkthrough of such existing tools and libraries which may benefit future research in this domain.

### 3.4.1. Codecrypt

Codecrypt is the first free-of-cost software package available for quantum-safe encryption and digital signatures. Codecrypt [68] was originally written by Miroslav Kratochvíl in 2013–2017, and is available in the Github repository (https://github.com/exaexa/codecrypt, accessed on 18 November 2020). The C++ language has been chosen for the development of this package. It uses McEliece Cryptosystem with IND CCA2 secure QD—MDPC variant for encryption and Hash-based Merkle Tree Algorithm (FMTSeq variant) for digital signatures. It has a command-line interface which provides multiple commands under the following categories:

- Encrypt, decrypt, sign and verify data
- Key management operations
- Input and output from standard I/O, as well as files
- View options including help and ASCII formatting

The cryptographic primitives of Codecrypt are similar to GnuPG software. The advantage of Codecrypt is that the complexity of all these quantum-safe cryptographic operations was scaled down to linear complexity from the respective exponential complexities in GnuPG. The speed of operations has increased significantly in Codecrypt due to this factor. The disadvantage of this software is that, although the signature schemes perform better than GnuPG, the encryption timing seems to be varying when the size of the message increases, except for the key generation part.

### 3.4.2. Open Quantum Safe

Open Quantum Safe (OQS) is a project for implementing and evaluating various quantum-resistant cryptographic primitives. It is created by a team headed by researchers Douglas Stebila and Michele Mosca [69]. The entire project is bifurcated into an open-source C library named 'liboqs' and a prototype integration suite of quantum-resistant algorithms in existing protocols and applications. The 'liboqs' library (https://github.com/open-quantum-safe/liboqs, accessed on 18 November 2020) provides the open-source implementation of various post-quantum key encapsulation mechanisms (KEM) and digital signature algorithms through a common API which can build on Windows, Linux and macOS. These can be executed overwork multiple platforms, including Intel x86, AMD, and ARM. The latest version of this library is 'liboqs 0.6.0' updated in June 2021, which is available in the Github repository, and is open to contributions. It includes all algorithms which made their entry to NIST Round 3 finalists or alternate candidates, except GeMSS. 'liboqs' can be integrated into different languages using language wrappers available with the package.

Prototype integration is achieved by modifying original protocols and applications to provide post-quantum primitives. The developers forked OpenSSL and BoringSSL to integrate liboqs supporting quantum-safe hybrid key exchange and authentication (KEA)

and cipher suites in TLS protocol. It also supports post-quantum algorithms in X.509 certificate generation and S/MIME/CMS message handling. Test servers are provided to test the interoperability of these prototypes. OpenSSH is also modified into OQS-OpenSSH for implementing KEA. New algorithms can be added to this suite at any time through the options available in the interface.

### 3.4.3. jLBC

The Java Lattice-based Cryptography Library (jLBC) (http://gas.dia.unisa.it/proje cts/jlbc/buildHowto.html, accessed on 17 November 2020) is a free Java library created by Angelo De Caro (https://github.com/adecaro/jlbc, accessed on 17 November 2020), a researcher at IBM Research Zurich. It is released under GNU public license. It has several modules providing API for the implementation of some lattice-based cryptosystems. 'jlbc-crypto' is the main supporting library implementing such algorithms. It utilizes the Bouncy Castle Cryptographic API framework (https://www.bouncycastle.org/, accessed on 18 November 2020), which acts as an intermediary providing Java Cryptography Extension (JCE) and Java Cryptography Architecture (JCA) (https://docs.oracle. com/javase/7/docs/technotes/guides/security/crypto/CryptoSpec.html, accessed on 18 November 2020) services. JCA provides a platform-independent, interoperable, and extensible architecture, and a set of APIs for implementing various cryptographic primitives. JCE supports advanced cryptographic operations. It also contains generators/processors for integration into other protocols and applications as well. All classes belonging to JCA and JCE are called engines. JCA engines are located in the 'java.security' package, whereas the JCE classes are located in the 'javax.crypto' package. The implementations of cryptographic algorithms are done via provider classes for these libraries. jLBC also implements few Homomorphic Encryption Schemes with the help of the Bouncy Castle framework (https://www.bouncycastle.org/java.html, accessed on 18 November 2020).

### 3.4.4. Microsoft's Lattice Cryptography Library

Microsoft product LatticeCrypto (https://www.microsoft.com/en-us/research/p roject/lattice-cryptography-library/, accessed on 20 November 2020) is a post-quantum cryptographic software that concentrates on lattice-based algorithms. It is created by a team of researchers including Michael Naehrig and Patrick Longa as principal contributors. The software is portable across a variety of platforms and is interoperable. It implements lattice-based cryptographic primitives for key exchange and authentication KEA built on R-LWE for its security. The ring-based Learning With Errors (R-LWE) Problem is unbreakable with quantum computers, and thus the cryptographic schemes in this software are supposed to be efficient against resisting quantum attacks. The implementation includes novel techniques for computing the number theoretic transform to achieve higher performance. The library successfully resists all timing and cache attacks reported so far. It supports Windows and Linux, and can be used on a wide range of platforms, including x86, x64, and ARM, and provides at least 128 bits of classical and quantum security.

### 3.4.5. libPQP

libPQP (https://github.com/grocid/libPQP/blob/869dfbf86b8fe3a56ba3dbddc86f 1291ee0263d1/README.md, accessed on 20 November 2020) is a Python post-quantum library that stands for post-quantum PGP. The source code of this library has not yet been audited and can be used only for testing purposes, as declared by the developer. It concentrates on improving the efficiency of the QC-MDPC variant of the McEliece cryptosystem and has several vulnerabilities. The final product was supposed to use Salsa-20 and Poly 1305 cryptographic primitives. The current version of the library was deprecated due to a key recovery attack10 on QC-MDPC variant published in 2016.

Besides these, there are other packages like ISARA Radiate Quantum-Safe Library that give a free sample code license, Palisade Homomorphic Encryption Library (https://palisade-crypto.org/, accessed on 20 November 2020) which is an open-source

lattice crypto-library, and Lattigo, which is a Go module that implements ring-learning-with-errors-based homomorphic-encryption primitives and multiparty-homomorphic-encryption-based secure protocols.

The implementation of post-quantum cryptographic schemes on the Android platform is highly necessary, as 90% of the mobile devices work on it. The liboqs, jLBC, and LatticeCrypto are all capable of adapting the algorithms to the ARM platform, and are efficient candidates for developing quantum-safe schemes for Android devices. 'liboqs' library is mostly used by developers for building such applications. The source code can be compiled and integrated into several applications via cross-platform compilers and APIs. A licensed library of quantum-safe cryptographic schemes has been created by O.S.Estrada and team14, which is a ready-to-use package for quantum-safe development in an Android environment. Moreover, numerous independent researchers are working to integrate quantum safety on android devices.

## 4. Code-Based Cryptography

Linear codes [19] are originally used for digital communication and are based on coding theory. Coding theory is an important study that attempts to minimize data loss due to errors introduced in transmission from noise, interference, or other forces. Data to be transmitted are encoded by the sender as linear codes which are decoded by the receiver. Data encoding is accomplished by adding additional information to each transmitted message, to enable the message to be decoded even if errors occur.

Different codes are being studied to provide solutions to various problems occurring in applications. The most prominent type of error-correcting code is called linear code. The linear codes can be represented by k x n matrices, where k is the length of the original messages, and n is the length of the encoded message. It is computationally difficult to decode messages without knowing the underlying linear code. This hardness underpins the security of the code-based cryptosystem, which includes all cryptosystems, symmetric or asymmetric, whose security relies, partially or totally, on the hardness of decoding in a linear error-correcting code, possibly chosen with some particular structure or in a specific family of linear codes.

Linear codes [70,71] are linear block codes over an alphabet $A = F_q$, where $F_q$ denotes the finite field with $q = p^l$ elements $l \in N^x$, $p$ prime. The alphabet is often assumed to be binary, that is $p = 2$, $l = 1$, $q = 2$, $F_2 = \{0,1\}$. The encoding of the source bits is done in blocks of predefined length k, giving rise to the name "block code".

The following are the matrices used in code-based cryptography.

- A generator matrix $G$ of an $[n,k]$ code $C$ is a $kn$ matrix $G$ such that $C = \{xG : x \in F_2^k\}$. Generator matrix is of the form $(I_k \mid Q)$, where $I_k$ is the $(k \times k)$ identity matrix and $Q$ is a $k \times (n\text{-}k)$ matrix (redundant part).
- A parity-check matrix $H$ of an $[n,k]$ code $C$ is an $(n\text{-}k) \times n$ matrix $H$, such that $C = \{c : \in F_2^n : Hc^T = 0\}$.
- Parity-check matrix $H$ is generated from the generator matrix as $H = (Q^T \mid I_{n-k})$.

Encoding process applies an injective $F_2$–linear function $f_c : F_2^k \to F_2^n$ on an input block of length $k$, i.e., every codeword can be generated by multiplying a source vector $x \in F_2^k$ with $GC = x \cdot G \mid x \in F_2^k \leq F_2^n$. Hence, the matrix $G$ corresponds to a map $F_2^k \to F_2^n$ mapping a message of length $k$ to an n-bit string. This encoding process corresponds to encryption in code-based cryptography.

The decoding process is about finding the closest codeword $x \in C$ to a given $y \in F_2^n$, assuming that there is a unique closest codeword. Decoding a generic binary code of length $n$ and without knowing anything about its structure requires about $2^{\frac{(0.5+o(1)) \times n}{log_2(n)}}$ binary operations, assuming a rate $\approx 1/2$. The following are the common decoding techniques [71]:

- List Decoding—Given $C$ and $x$, outputs the list $L_c(x,t) := \{c \in C \mid d(x,c) \leq t\}$, of all codewords at distance at most t to the vector $x$ with decoding radius t.

- Minimum Distance Decoding—Minimum distance decoding (MDD) is also known as nearest neighbour decoding, and tries to minimize the Hamming distance $d(x; y)$ for all codewords $y \in C$ given a received $F_2^n$.
- Maximum Likelihood Decoding—Given a received codeword $x \in F_2^n$, maximum likelihood decoding (MLD) tries to find the codeword $y \in C$ to maximize the probability that x was received, given that y was sent.
- Syndrome Decoding—For an $[n; k; d]$ code $C$, we can assume that the parity-check matrix $H$ is given. Syndrome $S = [Y][H^T]$, $Y$ is received code and $x = e + Y$ where $e$ is the error bit.

The decoding process corresponds to the decryption in code-based cryptography.

### 4.1. Different Types of Error-Correcting Codes

Error-correcting codes could be broadly classified into block codes and convolutional codes [71].

In block codes, the input is divided into blocks of $k$ digits. The coder then produces a block of $n$ digits for transmission, and the code is described as an $(n, k)$ code, as depicted in Figure 1. Linear block codes and non-linear block codes are types of block codes.

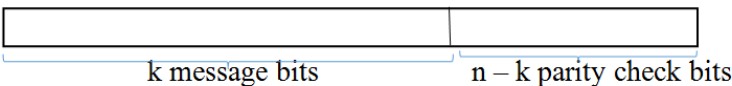

k message bits　　　　　n − k parity check bits

**Figure 1.** Sample Code Word of length n.

In convolutions coding, the coder input and output are continuous streams of digits. The coder outputs $n$ output digits for every $k$ digits input, and the code is described as a rate $k/n$ code.

### 4.2. Operations on Codes

In many applications, the allowed length of the error control code is determined by system constraints unrelated to error control. When the length of the code that one wishes to use is unsuitable, the code's length can be modified by the following operations on codes viz. (i) puncturing, (ii) extending, (iii) shortening, (iv) lengthening, (v) expurgating, or (vi) augmenting, which can be carried out in the following ways:

- An $(n, k)$ code is punctured by deleting any of its parity bits to become a $(n - 1, k)$ code.
- An $(n, k)$ code is extended by adding an additional parity bit to become a $(n + 1, k)$ code.
- An $(n, k)$ code is shortened by deleting any of its information bits to become a $(n - 1, k - 1)$ code.
- An $(n, k)$ code is lengthened by adding an additional information bit to become a $(n + 1, k + 1)$ code.
- An $(n, k)$ code is expurgated by deleting some of its codewords. If half of the codewords are deleted such that the remainder form a linear subcode, then the code becomes a $(n, k - 1)$ code.
- An $(n, k)$ code is augmented by adding new codewords. If the number of codewords added is 2k such that the resulting code is linear, then the code becomes a $(n, k + 1)$ code.

### 4.3. Properties to Be Fulfilled by Linear Codes

The properties to be fulfilled by linear codes depend on the code metric based upon which they are designed. Hamming metric, Rank metric and Lee metric codes are the most widely used codes. The description below provides an overview of each of these metrics.

#### 4.3.1. Hamming Metric

The Hamming metric is based on the Hamming Distance which is described below [72].

- The Hamming Distance between two linear codes in $F_2^n$ is the number of coordinates where they differ.
- The Hamming Weight of a linear code is the number of non-zero coordinates.
- The Minimum Distance $d_{min}$ of a linear code $C$ is the smallest Hamming weight of a nonzero codeword in $C$.
- A code is called maximum distance separable (MDS) code when its $d_{min}$ is equal to $n - k + 1$.

There exist many bounds for linear codes, which are mentioned below. Plotkin bound and the Hamming bound are upper bounds on $d_{min}$ for a given fixed value of $n$ and $k$. The Hamming bound is a tighter bound for high rate codes, but the Plotkin bound is for low rate codes.

- Singleton bound [70,72]: The minimum distance $d_{min}$ for an $(n, k)$ Binary Linear Block Code is bounded by $d_{min} \leq (n - k + 1)$.
- Plotkin bound [72]: For any $(n, k)$ binary linear block code, $d_{min} \leq \frac{n \times 2^{k-1}}{(2^k - 1)}$, the minimum distance of a code cannot exceed the average weight of all nonzero codewords.
- Gilbert Varshamov Bound [72]: For a fixed value of $n$ and $k$, Gilbert–Varshamov Bound gives a lower bound on $d_{min}$. According to this bound, if $\sum_{j=0}^{(d_{min}-2)} \binom{n-1}{j} \leq 2^{n-k}$ then there exists an $(n, k)$ binary linear block code whose minimum distance is outlast $d_{min}$.
- A Hamming sphere of radius $t$ contains all possible received vectors that are at a Hamming distance less than $t$ from a code word. The size of a Hamming sphere for an $(n, k)$ Binary Linear Block Code is, $V(n, t)$, where $V(n, t) = \sum_{j=0}^{t} \binom{n}{j}$.
- The Hamming bound: A t-error correcting $(n, k)$ Binary Linear Block Code must have redundancy n − k such that $(n\text{-}k) \geq \log_2 V(n, t)$. An $(n, k)$ Binary Linear Block Code which satisfies the Hamming bound is called a perfect code.

The Hamming metric codes which were used in code-based cryptography include [19] Reed–Muller Codes, Generalized Reed–Solomon Codes, Binary Goppa Codes, BCH Codes, Generalized Shrivastava Codes. More recently, to reduce the key size, some new kinds of codes with additional structures viz. Quasi-Cyclic Medium Density Parity Check Codes, Quasi-Cyclic Low Density Parity Check Codes, Quasi-Dyadic Generalized Shrivastava codes, Quasi-Cyclic Goppa Codes have been used for code-based cryptography.

4.3.2. Rank Metric

In coding theory [73,74], the most common metric is the Hamming metric, where the distance between two codewords is given by the number of positions in which they differ. An alternative to this is the rank metric. Let $X^n$ be an n-dimensional vector space over finite field $GF(q^N)$, where q is a power of prime and N is a positive integer. Let $(u_1, u_2, u_3, ..u_n)$ with $u_i \in GF(q^N)$ be a base of $GF(q^N)$ as a vector space over the field $GF(q)$. Every element $x_i \in GF(q^N)$ can be represented as $x_i = a_{1i}u_1 + a_{2i}u_2 + a_{Ni}u_N$. Hence, every vector $\vec{x} = (x_1, x_2, x_3, ...x_n)$ over the field $GF(qN)$ can be written as a matrix.

Every vector $\vec{x}$ over the field $GF(q^N)$ is a rank of the corresponding matrix $A(\vec{x})$ over the field $GF(q)$ denoted by $r(\vec{x}; q) \cdot (\vec{x}, \vec{y}) = r(\vec{x} - \vec{y}; q)$. A set $(x1, x2, x3, ...xn)$ of vectors from $X^n$ is called a code with distance $d = min(x_i, x_j)$. If the set also forms a k-dimensional subspace of $X^n$, then it is called a linear $(n, k)$ code with distance d. Such a linear rank metric code always satisfies the Singleton bound $d \leq n - k + 1$. Such codes are called maximum rank distance (MRD) codes.

$$\vec{x} = \left\| \begin{array}{ccc} a_{1,1} & a_{1,2} & a_{1,n} \\ a_{2,1} & a_{2,1} & a_{2,n} \\ a_{N,1} & a_{N,1} & a_{N,n} \end{array} \right\|$$

Non-binary linear codes have been defined using rank-metric. Such codes are called rank codes or Gabidulin codes. The rank metric defines a systematic way of building codes that could detect and correct multiple random rank errors. By adding redundancy with

coding k-symbol word to an-symbol word, a rank code can correct any errors of rank up to $t = [(d-1)/2]$, where d is a code distance. Some rank metric codes using code-based cryptography are [19] Rank Gabidulin Codes, Low Rank Parity Check Codes, Rank Quasi-Cyclic Codes, Ideal LRPC Codes.

### 4.3.3. Lee Metric

Lee metric was introduced by Lee as an alternative to Hamming metric for certain noisy channels [75]. Let $\mathbb{Z} = q\{0, 1, \ldots q - 1\}$ be the set of representatives of the integer equivalence classes modulo $q$. The Lee weight of any element $a \in \mathbb{Z}_q$ is given by $W_L(a) = min\{a, q - a\}$. Given an element $c = (c_1, c_2, c_3 \ldots c_n) \in \mathbb{Z}_q$, the Lee weight of c denoted as $W_L(c)$ is given by

$$W_L(c) = \sum_{i=1}^{n} \min\{a, q - a\}$$

For $c, e \in Z_q^n$, the Lee distance $d_L(c, e)$ between $c$ and $e$ is defined to be the Lee weight of their difference

$$d_L(c, e) = W_L(c - e) = \sum_{i=1}^{n} \min\{c_i - e_i(\bmod q), e_i - c_i(\bmod q)\}$$

The codes defined using Lee metric are called Lee codes. A Lee code will be specified by $(n, k, d_L)_q$. If $q = 2$ or 3, then the Hamming Metric and Lee Metric are identical. The Preparata Code and Kerdock Code are examples of Lee Codes. However, they have not been used in code-based cryptography.

### 4.4. Relationship between Codes

The relationship between codes has been abstractly specified in [19], as shown in Figure 2.

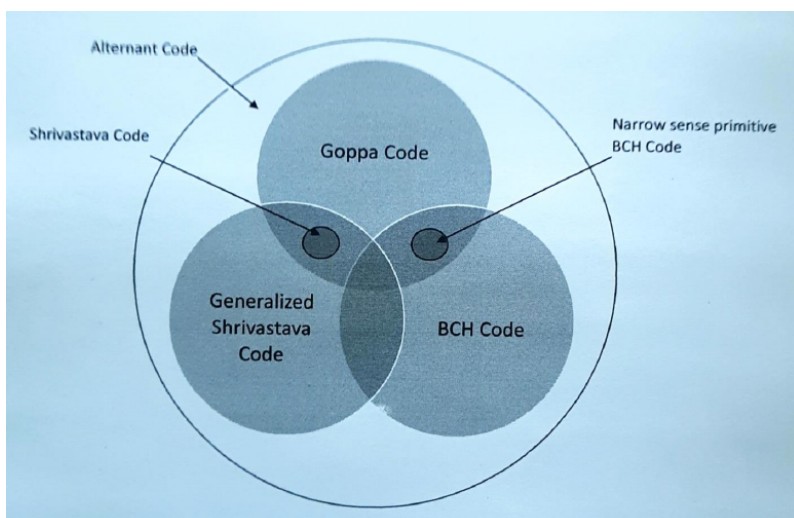

**Figure 2.** Relationship between codes.

Based on our study of the various linear codes, the following relationship between codes has been identified as part of this research work, which is depicted in Figure 3. The relationship between the special codes is depicted in Figure 4.

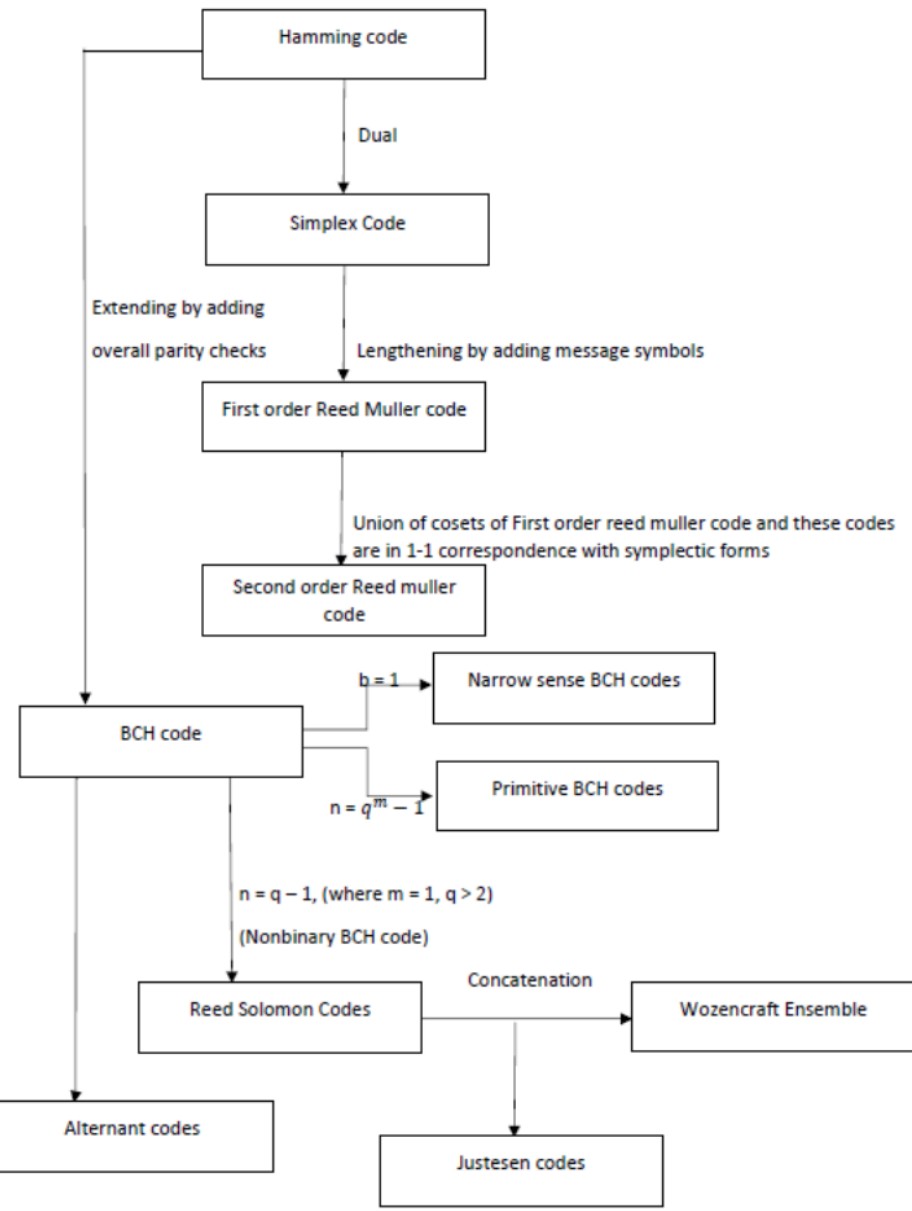

**Figure 3.** Relationship Between Codes.

*4.5. Common Code-Based Cryptographic Algorithms*

Research in code-based cryptography provides for public-key encryption and digital signature algorithms. An overview of the same is provided in subsequent sections.

4.5.1. Code-Based Encryption

McEliece and Niederreiter are two of the earliest cryptographic algorithms developed in code-based cryptography. The McEliece was initially built in 1978 using the binary goppa code with [n,k] = [1024, 524] [76]. Subsequently, several variants of McEliece were built using different linear codes [26]. However, those variants were proven to be susceptible to attacks [19,77], and only the McEliece built using the Binary Goppa Code is found to be quantum attack resistant to date. Thus, it has also been chosen for the third round of standardization by NIST [24]. The McEliece has quadratic complexity in block length, and no polynomial-time quantum algorithm is found to decode the general linear block code [78]. Moreover, the algorithm incorporates an element of randomness in every encryption by the use of e, which is a randomly generated error vector [79]. These are the advantages of McEliece [79]. The large key size is the limitation of McEliece [78,79].

The Niederreiter cryptographic algorithm [80] was developed in 1986, and is very similar to McEliece, and encrypts messages using parity check matrices, unlike generator matrices used in McEliece, and uses the Generalized Reed–Solomon Codes. Table 5 shows an indicative comparison of the two cryptosystems.

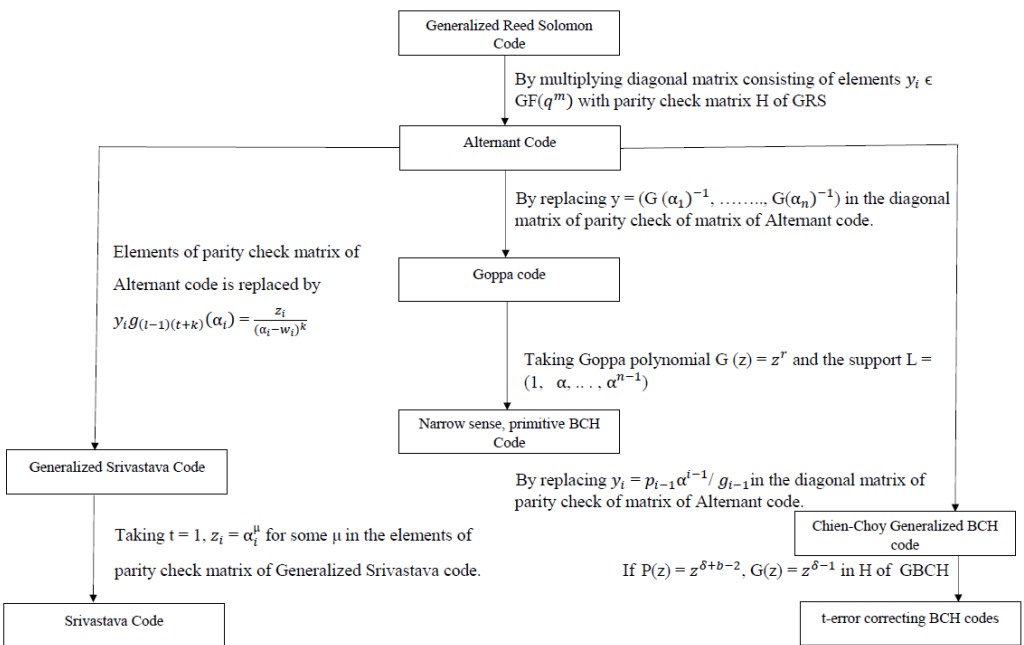

**Figure 4.** Special codes.

**Table 5.** Common Code-based Cryptosystems, Codes used and thier Application.

| Serial Number | Code-Based Cryptography | Technique & Codes Used | Applied to |
| --- | --- | --- | --- |
| 1 | Mc Eliece | Binary Goppa Codes, GRS, Concatenated Codes, Product codes, Quasi- Cyclic, Reed muller codes, Rank matric ( Gadidulin) codes, LDPC, MDPC, Genaralized Shri vastava codes | Computing Systems, Embedded Devices [81], FPGA systems [82] |
| 2 | Niederreiter | GRS Codes, Quasi-Cyclic Codes, Binary Goppa codes | Computing Systems, FPGA Systems [83] |

4.5.2. Code-Based Signature Schemes

Signature schemes based on linear codes have been developed based on the FDH-like (full domain hash) approach by Courtois–Finiasz–Sendrier (called CFS) [84], and uses Goppa Codes. The modified CFS signature—mCFS—was developed by Dallot [85]. Signature schemes based on Fiat–Shamir Transformation on zero-knowledge identification schemes have been developed by Stern et al. [86], Jain et al. [87], and Cayrel et al. [88]. However, none of the code-based signature schemes have been shortlisted by NIST for the third round of standardization. A comparison of the latter three signature schemes is provided in Table 6 [89].

In addition to these works, a signature scheme with the name RankSign in the rank metric setting was proposed [90]. The security of RankSign builds on the assumption that the special codes are indistinguishable from random linear rank metric codes. However, this scheme was attacked with a structural key-recovery attack in 2018 [91]. The Random Code-based Signature Scheme (RaCoSS) was submitted to NIST [92]. The RaCoSS scheme builds upon random linear codes. This scheme was attacked [93] two days after submission.

Subsequently, the scheme was patched [94] and attacked again in the same year—2018 [95]. The main issue of the schemes [92,94] is that the weight of valid signatures is large.

**Table 6.** Comparison of Code-based signature schemes.

|              | Stern      | Jain et al. | Cayrel et al. |
|--------------|------------|-------------|---------------|
| Keygen       | 0.0170 ms  | 0.0201 ms   | 0.339 ms      |
| Sign         | 31.5 ms    | 16.5 ms     | 24.3 ms       |
| Verify       | 2.27 ms    | 135 ms      | 9.81 ms       |
| sk           | 1.24 bits  | 1536 bits   | 1840 bits     |
| pk           | 512 bits   | 1024 bits   | 920 bits      |
| System prams | 65.5 kB    | 65.5 kB     | 229 kB        |
| Signature    | 245 kB     | 263 kB      | 229 kB        |

In the same year of 2018, Persichetti [96] conformed the Lyubashevsky scheme to random quasi-cyclic Hamming metric codes. Persichetti's proposal does not suffer from the weakness of RaCoSS, because the weight of a valid signature is below the GV bound of the code [97]. This scheme [96] was attacked in the two subsequent independent works [98,99].

In 2019, Anguil et al. [100] proposed a signature scheme with the name Durandal in the rank metric context. The security of Durandal builds on a complicated new problem—PSSI+. It is not proven that the signature distribution is independent of the secret key.

The Wave [101] signature scheme was proposed in 2019, which follows the hash-and-sign framework. The security of Wave builds on the new assumption that generalized $(U, U + V)$ codes are independent of random linear codes. The hardness of distinguishing generalized $(U, U + V)$ codes from random linear codes is still unclear [97]. The output signature is proven to be independent of the secret key. Different from traditional code-based cryptography, the Wave signature chooses the desired weight of the underlying syndrome decoding problem to lie at the large end [97].

In summary, the existing works on secure code-based signature schemes build the security on immature intractability assumptions.

### 4.6. Attacks in Code-Based Cryptography

The attacks on code-based cryptography can be classified into critical [102] and non-critical attacks (Chapter 4 in [103]).

Critical Attacks:

The following are the different types of critical attacks possible in code-based cryptography. Critical attacks aim to recover the key or cryptanalysis of the ciphertexts to decrypt the ciphertext.

1. **Broadcast attack:** This attack [104] aims to recover a single message sent to several recipients. Here, the cryptanalyst knows only several ciphertexts of the same message. Since the same message is encrypted with several public keys, it was found that it is possible to recover the message. This attack has been used to break the Niederreiter and HyMES (Hybrid McEliece Encryption Scheme) cryptosystems.
2. **Known partial plaintext attack:** A known partial plaintext attack [105] is an attack for which only a part of the plaintext is known.
3. **Sidelnikov–Shestakov attack:** This attack [106] aims to recover an alternative private key from the public key.
4. **Generalised known partial plaintext attack:** This attack [102] allows to recover the plaintext by knowing a bit's positions of the original message.
5. **Message-resend attack:** A message-resend condition [102] is given if the same message is encrypted and sent twice (or several times) with two different random error vectors to the same recipient.

6. **Related-message:** In a related-message attack against a cryptosystem, the attacker obtains several ciphertexts such that there exists a known relation between the corresponding plaintexts.

7. **Chosen plaintext (CPA):** A chosen-plaintext attack [102] is an attack model for cryptanalysis which presumes that the attacker can choose arbitrary plaintexts to be encrypted and obtain the corresponding ciphertexts. The goal of the attack is to gain some further information that reduces the security of the encryption scheme. In the worst case, a chosen-plaintext attack could reveal the scheme's secret key. For some chosen-plaintext attacks, only a small part of the plaintext needs to be chosen by the attacker: such attacks are known as plaintext injection attacks. Two forms of chosen-plaintext attack can be distinguished: *Batch chosen-plaintext attack*, where the cryptanalyst chooses all plaintexts before any of them are encrypted.
   *Adaptive chosen plaintext attack*, where the cryptanalyst makes a series of interactive queries, choosing subsequent plaintexts based on the information from the previous encryptions.

8. **Chosen-ciphertext attack (CCA)** In a chosen-ciphertext attack [102], an attacker has access to a decryption oracle that allows decrypting any chosen ciphertext (except the one that the attacker attempts to reveal). In the general setting, the attacker has to choose all cipher texts in advance before querying the oracle.
   In the adaptive chosen-ciphertext attack, formalized by Rackoff and Simon (1991), one can adapt this selection depending on the interaction with the oracle. An especially noted variant of the chosen-ciphertext attack is the lunchtime, midnight, or indifferent attack, in which an attacker may make adaptive chosen-ciphertext queries, but only up until a certain point, after which the attacker must demonstrate some improved ability to attack the system.

9. **Reaction attack:** This attack [107] can be considered as a weaker version of the chosen-ciphertext attack. Here, instead of receiving the decrypted ciphertexts from the oracle, the attacker only observes the reaction of this one. Usually, this means whether the oracle was able to decrypt the ciphertext.

10. **Malleability attack:** A cryptosystem is vulnerable to a malleability attack [107] of its ciphertexts if an attacker can create new valid ciphertexts from a given one, and if the new ciphertexts decrypt to a clear text which is related to the original message.

Non-Critical Attacks:

These kinds of attacks attempt to decode a ciphertext by solving either the general decoding problem or the syndrome decoding problem, and do not focus on recovering the secret key.

1. **Information Set Decoding (ISD) Attack:** ISD algorithms [103]) are the most efficient attacks against the code-based cryptosystems. They attempt to solve the general decoding problem. That is, if m is a plaintext and $c = mG + e$ is a ciphertext, where e is a vector of weight t and G a generator matrix, then ISD algorithms take $c$ as input and recover $m$ (or, equivalently, $e$).

2. **Generalized Birthday Algorithm (GBA):** The GBA algorithm [108] is named after the famous birthday paradox which allows one to quickly find common entries in lists. This algorithm tries to solve the syndrome decoding problem. That is, for given parity check matrix $H$, syndrome s, and integer t, it tries to find a vector e of weight t such that HeT = s T. In other words, it tries to find a set of columns of H whose weighted sum equals the given syndrome.

3. **Support Splitting Algorithm (SSA):** The SSA algorithm [109] decides the question of whether two given codes are permutation equivalent, i.e., one can be obtained from the other by permuting the coordinates. For this, SSA makes use of invariants and signatures (not digital signatures). An invariant is a property of a code that is invariant under permutation, while a signature is a local property of a code and one of its coordinates. The difficulty of using invariants and signatures to decide whether

two codes are permutation equivalent is that most invariants are either too coarse (i.e., they take the same value for too many codes which are not permutation equivalent) or the complexity to compute them is very high. The SSA solves this issue by starting with a coarse signature and adaptively refines it in every iteration.

*4.7. Related Work*

This section provides an overview of the various surveys that have been conducted in CBC. This has been provided to substantiate the fact that the prospective research dimensions to be explored that have been delineated by us in the subsequent section have not been in the focus of any of the existing works in CBC. Towards this, we reviewed the surveys from 2008 till now. In 2008, Overbeck and Sendrier [110] published a comprehensive state-of-the-art of code-based cryptography. In this work, the authors illustrate the theory and the practice of code-based cryptographic systems. In 2011, Cayrel et al. [25] presented a survey paper which has included state-of-the-art of publications since 2008 in code-based cryptography, including encryption and identification schemes, digital signatures, secret-key cryptography, and cryptanalysis. This work provides a comprehensive study and an extension of the chapter "Code-based cryptography" of the book [111].

In the same year, 2014, Repka and Cayrel [112] proposed a survey paper on cryptography based on error-correcting codes. This paper surveys the code-based cryptography existing schemes, as well as implementations and side-channel attacks. This work also recalls briefly the basic ideas and provides a roadmap to readers.

In 2015, PQCRYPTO project Horizon 2020 ICT-645622 [113] published a report on post-quantum cryptography for long-term security which provides the PQCRYPTO project's intermediate report on optimized software. This report also provides the preliminary software implementation results of selected post-quantum schemes and the corresponding parameters for embedded systems. This report surveys modern post-quantum schemes regarding their implementation on such small embedded microcontrollers. This report reviews the most popular schemes in post-quantum cryptography such as encryption and digital signatures schemes.

In 2017, Sendrier [26] published a survey paper that focuses on the McEliece public-key encryption scheme and its variants which are the candidates of post-quantum public-key encryption standard. This paper also focuses on other cryptographic primitives using codes such as zero-knowledge authentication and digital signature.

In parallel to this work in 2017, Bucerzan et al. [114] analyzed the evolution of the main encryption variants coming from code-based cryptography field. The authors focus on security issues and Rank-based cryptography. This paper provides the details and survey on the McEliece encryption scheme. In addition to this, these papers detail the main security threats for the scheme and for each of the mentioned variants.

The most recent survey in code-based cryptography was published in 2018 [115]. In this paper, the authors survey code-based cryptography, essentially for encryption and signature schemes. The authors also provide the main ideas for theoretical and physical cryptanalysis.

According to the best of our knowledge, these are the surveys available in the direction of code-based cryptography research. A review of these surveys helped to understand that the existing works in code-based cryptography have mostly identified the following as the prospective research dimensions that could be explored in CBC research:

- reduction of key size—large key size is one of the important limitations of CBC and reducing the key size is an important research direction explored
- use of new kinds of linear and non-linear codes in CBC, viz. QC-MDPC, QC-LDPC, etc.—recently CBC using these kinds of codes have been proposed to overcome various kinds of attacks
- algorithms for resolving new kinds of security attacks—there are various security attacks possible in CBC and various techniques and algorithms to counteract the same have been proposed

- evolving new signature schemes—signature schemes using CBC were a recent addition to CBC research

The above research directions proposed in the existing works help to emphasize the fact that the ones which have been provided in this paper are novel and yet to receive the interest of the CBC research community.

## 5. Research Directions Identified in Code-Based Cryptography

In this section, we lay out some of the research directions which have been least explored and remain as white spaces in the code-based cryptographic research. Though this paper elaborates on both PQC and code-based cryptography, the future research directions confine only to code-based cryptography for two reasons (i) future research direction in PQC ultimately boils down to any one of the PQC schemes viz. code-based, lattice-based, etc and (ii) our current research directions centers around code-based cryptography.

### 5.1. Dynamic Code-Based Cryptographic Algorithms

The linear codes are many in number and various code-based cryptographic algorithms using these code variants have been proposed. However, these cryptosystems except the McEleice cryptosystem which uses binary Goppa codes have been reported to be broken, discouraging the use of other linear codes. Even The variants of the McEliece algorithm using the other different types of linear codes apart from binary Goppa codes are susceptible to attacks. This is because the static code used in the algorithm is known earlier and also it results in a known structure of the linear code which could be cryptanalysed easily. However, the study of linear codes and the relationships between them, as described above explicate that it is possible to transform one code to another utilizing some operations on codes viz. augmenting, puncturing, extending, ... etc as mentioned in Section 4.2. The existing variants of code-based cryptographic algorithms, for example, the McEliece uses a single code (binary goppa) as the basis for the encryption algorithm. Since, it is possible to transform one linear code to another using the possible code transformation operations, the same could be exploited in the encryption. Thereby, the cryptographic algorithm can dynamically choose to use any type of linear code to perform the encryption operation. This dynamic code transformation may produce any other existing linear codes or a new code that fulfills the properties for linear codes viz. Gilbert Varshmov bound, Singleton bound, etc. – for example, from the alternant code one can transform to the Generalized Shrivastava code or a new code fulfilling the linear code bounds. This dynamic approach provides two-fold advantages viz. (i) the cryptographic algorithm can dynamically choose to use a particular type of linear code randomly with every session or even in between sessions so that it becomes very difficult to break the cipher since the structure of the linear code keeps varying (ii) renders the otherwise unsafe linear codes to provide for quantum attack resistance thereby augmenting the utility of the various types of linear codes available in code-based cryptographic algorithms.

All aspects of this proposal including the study of various relationships between codes, (in this paper we have provided the relationship between only a subset of linear codes) the transformations that can help a linear code to transform into another code dynamically, the design and engineering of such cryptographic algorithms efficiently are the research directions intended to be explored here.

### 5.2. Use of Other Types of Codes in Code-Based Cryptography

Codes have existed for a long in the computing domain, and there are a variety of codes available, which have been used for a variety of encoding purposes. The following is a representative list of encoding purposes commonly encountered in the computing domain [116]. (i) To encode data for digital data communication (ii) To encode data for digital data communication with error correction capabilities (iii) To encode data in a compressed format for faster message communication (iv) To represent data in a digital system (v) To store and manipulate data in a digital system (vi) Programmatic

representation of Character set (vii) To communicate digital data confidentially (viii) To represent data or data set features to be used in machine learning (ix)To represent data in a format comprehensible for visually challenged persons.

In each of the above said seven purposes, the encoding achieves either one of the following encoding (i) Alphabets/character set encoded to another alphabet (ii) Alphabets/character set encoded to a sequence of alphabets (iii) character set encoded to a number (iv) Alphabets/character set encoded to binary/BCD/Hexadecimal/Octal through ordinal encoding, (v) Alphabets/character set encoded to image(s)/symbol(s)/pattern(s) (vi) Alphabets/character set encoded to a compressed code (vii) Binary data encoded to linear codes (viii) Data encoded to a vector (ix) Data encoded to a non-numerical label.

As is evident from above, the said encoding types work at multiple levels of abstractions viz. The encoding may work to encode alphabets or a complete character set, or it may encode binary data, or it may encode a data unit.

Since we consider encoding from a cryptographic perspective, the following requirements are to be fulfilled by the code/encoding technique to constitute a complete and secure code/encoding technique. These requirements have already been identified in our earlier work in [117].

*Support for encoding of complete character set*—Some coding techniques provide code for encoding only the alphabets like the Caesar cipher, A1Z26, Atbash codes, etc. However, since a plain text message may be alphanumerically comprising of alphabets, numbers, and special characters, the coding technique must help to encode the complete character set.

*Possible for representation, manipulation, and storage in digital systems*—Some encoding techniques result in the production of codes which are in the form of an image(s) or symbol(s) or pattern(s) for the alphabet or character set that is encoded. However, these image(s) or symbol(s) or pattern(s) cannot be directly represented, manipulated, and stored in digital systems. Dorabella cipher, Morse codes, Dice Cipher, Rosicrucian Cipher are some examples of such coding techniques which produce symbols or patterns as codes that are not suitable for direct representation, manipulation, and storage in digital systems.

*Enables Data Hiding*—Since the codes are approached from the perspective of cryptography, enabling data hiding as a result of encoding is one of the important properties expected to be fulfilled by the code and hence the encoding technique.

*Support for multiple data type*—Since, the plain text may not only be plain text and also be of any data type viz., audio, video, etc. it is essential that the encoding technique can encode any data type.

*Dynamicity of encoding*—Usually, the encoding techniques provide for static encoding whereby a particular element of a character set or data is always encoded to the same code any time encoding happens. This may enable ease of encoding – but approached from a cryptographic perspective, a dynamic encoding scheme which produces a different code for an element of the character set or data for different data communications sessions should help augment the strength and effectiveness of the encoding.

*Variable Encoding*—Even in a single session of communication, the plain text to be encoded should be subject to the production of varied codes unlike the Caesar cipher codes, Atbash Cipher codes which produce the same code for the same character in all of the plain text which leaves a lot of ques for easy decoding.

*Hard Decoding*—The encoding technique should be able to produce codes which are hard to decode. In other words, the decoding should be a computationally hard problem to solve.

*Randomness of Encoding*—Usually, an encoding technique comprises steps constituting for the generation of code in some particular static order. If the steps of encoding itself are randomized, then it constitutes a higher strength of the generated code making the decoding a computationally hard problem to solve.

*Possibility for Decoding*—Though the decoding is required to be a computationally hard problem, it should be possible to decode the code given the key – unlike the hash codes which cannot be decoded since the process of hashing is a one-way function.

*Random Character Set and Collating Sequence*—In encoding, character set is considered and its collating sequence is usually adopted. To provide for a high degree of randomness and hence to render strength to the code, it should be possible to use alphabets across character sets and specify a user-defined collating sequence of those alphabets chosen.

Table 7 shows the various requirements for data encoding listed above and the types of codes available in each of them and the properties fulfilled by each of them.

From the comparison in Table 7, it is observed that more than the linear codes which are presently used in code-based cryptography, the DNA codes provide promising scope to be used for cryptography. This has been described in detail in [117]. Though research in DNA cryptography is active and the domain has been explored in interesting dimensions, DNA cryptography has not been proven to be quantum attack resistant. If DNA cryptography is proved to be quantum attack resistant, then it provides for a bio-inspired, best value addition to the field of code-based cryptography. This dimension needs to be explored in further detail.

Thus, DNA codes and identifying other kinds of codes amenable for code-based cryptography and designing and engineering code-based cryptographic algorithms using these other kinds of codes efficiently are prospective research directions to be explored.

### 5.3. Privacy-Preserving Code-Based Cryptography

Privacy-preserving encryption algorithms are the need of the hour owing to the growing privacy issues and concerns globally. Privacy-preserving encryption can be achieved in the following ways:

- Attribute-based Encryption—Key Policy-based encryption and Cipher Policy-based encryption [118]
- Homomorphic encryption [119]

Whereas attribute-based encryption provides for selective decryption of ciphertext based on the fulfillment of attributes by the receiver, homomorphic encryption enables one to perform computations on the encrypted data itself, eliminating the requirement for decryption.

In code-based cryptography, only the McEliece cryptosystem has been proven to be somewhat homomorphic [120]. However, attribute-based encryption in code-based cryptography is yet to be explored. Hence, there is a need for a lot of research to enable the maturity of this dimension of code-based cryptography and its practical use in privacy demanding domains such as healthcare, the financial sector, etc. This constitutes another prospective line of research in code-based cryptography.

### 5.4. Prospective Applicability of Codes with Lattice-Based Cryptography

Codes and Lattices are having similar mathematical properties. A linear code *C* of length *n* and dimension *k* is a k-dimensional subspace of finite field typically endowed with hamming metric.

Given n-linearly independent vectors $b_1, ..., b_n$ in $R^n$, the lattice generated by them is the set of vectors

$L(b1, ..., bn) = \sum_{i=1}^{n} xibi : xi \in Z for 1 \leq i \leq n$

The vectors $b_1, \ldots, b_n$ are known as a basis of the lattice.

**Table 7.** Requirements fulfillment of various codes.

| Serial Number | Purpose of Encoding | Code | Type of Code | Properties to Be Fulfilled for Encoding | | | | | | | | | Support |
|---|---|---|---|---|---|---|---|---|---|---|---|---|---|
| | | | | Complete Character Set Encoding | Representation, Manipulation and Storage in Digital Systems | Data Hiding | Support Multiple Data Types | Dynamic Encoding | Variable Encoding | Hard Decoding | Randomness of Encoding | Possible for Decoding | |
| 1 | To encode data for digital data communication | All codes generated using Line coding techniques [121] | Binary data encoded to digital signals | ✓ | ✓ | ✓ | ✓ | ✗ | ✗ | ✗ | ✗ | ✓ | ✗ |
| 2 | To encode data for digital data communication with error correction capabilities [121] | All codes generated using Block Codes and Convolutional Coding Techniques [72] | Binary data encoded to linear or non-linear codes with error detection and correction capabilities | ✓ | ✓ | ✓ | ✓ | ✗ | ✗ | ✓ | ✗ | ✓ | ✗ |
| 3 | To encode data in a compressed format for faster message communication | Huffman Codes [122], | Alphabets/character set encoded to a compressed code | ✓ | ✓ | ✓ | NA | ✗ | ✗ | ✓ | ✗ | ✓ | ✗ |
| | | Morse Code [123] | (a)Alphabets/ Character set encoded to a compressed code | ✓ | ✓ | ✓ | NA | ✗ | ✗ | ✓ | | ✓ | ✗ |
| | | | (b)Alphabets/ Character set encoded to image(s)/ symbols(s)/ patterns(s) | | | | | | | | | | |
| 4 | To represent data in a digital system | Ascii, Unicode,..[124,125] | Alphabets/character set encoded to a number | ✓ | ✓ | ✓ | ✗ | ✗ | ✗ | ✗ | ✗ | ✓ | ✗ |
| 5 | To store and manipulate data in a digital system | Binary, BCD, Hexadecimal, Octal [126] | Alphabets/character set encoded to binary/BCD/Hexadecimal/Octal through ordinal encoding | ✓ | ✓ | ✓ | ✓ | ✗ | ✗ | ✗ | ✗ | ✓ | ✗ |
| 6 | Programmatic Representation of Character Set | HTML Code [127] | Alphabets/character set encoded to a Hexadecimal number | ✓ | ✗ | ✓ | ✗ | ✗ | ✗ | ✗ | ✗ | ✓ | ✗ |
| 7 | To communicate digital data confidentially | Atbash, Caesar Cipher, Columnar Cipher, Combination cipher, Grid Transposition cipher, Keyboard Code, Phone code, Rot Cipher, Rout Cipher [116] | Alphabets encoded to another alphabet | ✗ | ✓ | ✓ | ✗ | ✗ | ✗ | ✗ | ✗ | ✓ | ✗ |
| | | A1Z26 | Alphabets encoded to a number | ✗ | ✓ | ✓ | ✗ | ✗ | ✗ | ✗ | ✗ | ✓ | ✗ |
| | | QR Code, Bar Code, Dice Cipher, Digraph cipher, Dorabella Cipher, Rosicrucian Cipher, Pigpen cipher [116,128,129] | Alphabets/Character Set encoded to image(s)/symbol(s)/pattern(s) | ✗ | ✓ | ✓ | ✗ | ✗ | ✗ | ✗ | ✗ | ✓ | ✗ |
| | | Francis Bacon Code [116] | Alphabets/character set encoded to a sequence of alphabets | ✗ | ✓ | ✓ | ✗ | ✗ | ✗ | ✗ | ✗ | ✓ | ✗ |
| | | DNA Code [117] | Alphabets/character set encoded to a sequence of alphabets | ✓ | ✓ | ✓ | ✓ | ✓ | ✓ | ✓ | ✓ | ✓ | ✓ |
| 8 | To represent data or data set features to be used in machine learning | Ordinal Code or label code | Data encoded to a non-numerical label | NA | ✓ | ✓ | NA | NA | NA | NA | NA | ✓ | ✗ |
| | | One Hot Encoding, Dummy Encoding, Effect Encoding, Binary Encoding, Base N Encoding, Multi-Label Binarizer, DictVectorizer [130] | Data is converted to a vector | NA | ✓ | ✓ | NA | NA | NA | NA | NA | ✓ | ✗ |
| | | Hash Encoding [130] | Data is converted to its hash value | NA | ✓ | ✓ | NA | NA | NA | NA | NA | ✗ | ✗ |
| | | Bayesian Encoding [131] | Data is encoded to its average value | NA | | ✓ | NA | NA | NA | NA | NA | NA | ✗ |
| 9 | To represent data in a format comprehensible for visually challenged persons | Braille Code [132] | Alphabets/character set encoded to image(s)/symbol(s)/pattern(s) | ✓ | ✓ | ✓ | NA | ✗ | ✗ | ✗ | ✗ | ✓ | ✗ |

Both of them are vector spaces over some finite fields.

1.  Typical lattice-based cryptographic schemes have used q-ary lattices to solve SIS and LWE problems [133]. Linear code of length n and dimension k is a linear subspace which is called a q-ary code. The possibility of using q-ary lattices [134] to implement ternary codes i.e., q-ary codes in code-based cryptographic schemes is an unexplored area. It may be noted here that DNA cryptography is a Quaternary code which has received due exploration from the authors but only needs to be ascertained for its quantum attack resistance.
2.  There is a major lattice algorithmic technique that has no clear counterpart for codes, namely, basis reduction. There seems to be no analogue notions of reduction for codes, or at least they are not explicit nor associated with reduction algorithms. We are also unaware of any study of how such reduced bases would help with decoding tasks. This observation leads to two questions.

- Is there an algorithmic reduction theory for codes, analogue to one of the lattices?
- If so, can it be useful for decoding tasks?

These questions are potential leads towards prospective research directions in code-based cryptography.

## 6. Conclusions

Post-quantum cryptography research has branched out in many dimensions and a considerable research outcome has been emerging in each of these dimensions. While this evinces the maturity of post-quantum cryptography research, each of these outcomes is available in discrete sources hindering the broad spectrum view and comprehension of these outcomes. This paper addresses this limitation, whereby, it provides a one-stop reference of the entire spectrum of post-quantum cryptography research and briefs the research happening in those directions.

Furthermore, from the NIST standardization, it has been observed that though code-based cryptography provides scope to be recognized as a complete cryptosystem with the availability of encryption, key exchange, and digital signature schemes, unlike its post-quantum counterparts which provide for a subset of these. Hence, an overview of the research directions that have been explored in code-based cryptography has been provided and the promising research directions that can augment the prospects of code-based cryptography from the codes perspective have been identified and described. Thereby, this paper provides two solid contributions in the roadmap of post-quantum computing research.

**Author Contributions:** Idea Conceptualization and Writing, C.B. and K.S.; methodology, C.B. and K.S.; formal analysis, C.B.; investigation, C.B. and K.S.; resources, C.B. and K.S.; writing—original draft preparation, C.B. and K.S.; writing—review and editing, C.B., K.S. and G.G.; supervision, C.B., K.S. and M.R.; project administration, K.S.; funding acquisition, K.S. All authors have read and agreed to the published version of the manuscript.

**Funding:** This research received funding from IRT SystemX.

**Institutional Review Board Statement:** Not applicable.

**Informed Consent Statement:** Not applicable.

**Data Availability Statement:** The data presented in this study are available in artcile.

**Acknowledgments:** This research work has been carried out under the leadership of the Institute for Technological Research SystemX, and therefore granted within the scope of the program "Recherche Exploratoire".

**Conflicts of Interest:** The authors declare no conflict of interest.

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
