# Peer review of "Post-Quantum and Code-Based Cryptography—Some Prospective Research Directions"

_cryptography, doi:10.3390/cryptography5040038_

Round 1

Reviewer 1 Report

The paper surveys about code-based cryptography which is a direction in post-quantum cryptography. The paper is well-written and organized. I hence suggest to accept the paper.

Author Response

We thank to the reviewer to suggest to accept our paper. 

Reviewer 2 Report

‎In this paper, the authors attempt to put forward general directions for research of code-based cryptography with post-quantum (PQ) security. It in fact makes a broader overview (in sections 1-4), bringing together several other, previously published surveys.   Summary: Cons: Although such a survey generally has merit, although it does not make concrete progress on the proposed direction, I have several other objections.   1. The nature of the survey is such that it presents numerous directions, providing little details on each. It seems like they try to include a lot of information, but there is little intuition and structure there. It is better to focus on fewer directions, on which you have somethin more concrete to say. In particular, the lack of concreteness makes it hard to assess  feasibility of the various directions, and raises concerns about the author's understanding of what is already there. For example, there is already vast literature dealing with Lattice based crypto, with the goal of PQ security in mind.   2. There are several imprecise statements. for example   Line 80 -  "ultimate in parallel computing, unclear and seems imprecise..". Fails to mention limitations and subtleties of QC Line 103 - "..need to evolve"  -it is not clear what tis the difference between the two states   Line 654 - The broadcast attack" on page 19, talks about an attack for recovering the message. Sounds quite strong.. Is it specific to a concrete encryption scheme?   3. The main contribution of the paper is proposing directions for code-based-crypto with resilience against quantum attackers aka post quantum crypto). Some of the directions are not clear:   1. You propose to research "dynamic" cryptographic primitives (which, BTW?), that rely on different underlying linear codes. With one of the goals being PQ security. If I understand correctly, the tradeoff is efficiency via PQ security. So, the high level idea is to understand what is more critical for a certain application, and decide what to use accordingly. The high level idea is clear. So, what aspects should the research focus on?! Engineering aspects of fully implementing such a system.    * On the positive side: the discussion of DNA based codes seems like a direciton the authors have considered somewhat deeper. It would be nice to see this and few other directions  elaborated further, in order to consider thet work for publication, in my opinion.

Author Response

Answer to comment 1: 

This paper provides an overview of the various research directions that have been explored in post-quantum cryptography and, in specific, the various code-based cryptography research dimensions that have been explored. Some potential research directions that are yet to be explored in code-based cryptography research from the perspective of codes is a key contribution of this paper.

The novelty of the paper lies in the elicitation of the prospective research directions that could be explored in code-based cryptography, which to our knowledge does not overlap with any of the state of art works.

Answer to comment 2:

This paper is not intended to be a survey paper in the first place.

However, we wanted to provide an overview of all the research directions which have been explored in code-based cryptography – the details of which can be found in the corresponding referred papers mentioned in our paper. Hence, we have not focused on detailing the existing research directions and stay confined to only providing the leads of the same.

Answer to comment 3:

Line 80 has been removed.

Line 103 – Currently lines 99-101 has been modified to make it clear.

Line 654 – currently line 651 – ‘the broadcast attack has been shown to break the Niederreiter and HyMES (Hybrid McEliece Encryption Scheme) cryptosystems’. This line has been included in the paper.

Answer to comment 4:

In research direction 1, We propose the use of different related linear codes for generating the cipher texts. Here, we have provided enough details based on our study of relationship between codes which is presented in section 4.4 which is why this could be a feasible approach for designing code-based cryptographic algorithms. This approach is yet to be used in code-based cryptography.

‘All aspects of this proposal including the study of various relationship between codes, (in this paper we have provided the relationship between only a subset of linear codes) the transformations that can help a linear code to transform into another code dynamically, the design and engineering of such cryptographic algorithms efficiently are  the research directions intended to be explored here’.

The above line has been included in the paper in line 815

In research direction 2 – using other kinds of codes for code-based cryptography, we have given considerable analysis with the necessary details to explain what are the various criteria to be fulfilled by a code being considered for designing code-based cryptographic algorithms and show how DNA codes are prospective for designing code-based cryptographic algorithms.

‘Thus, DNA codes and identifying other kinds of codes amenable for code-based cryptography and designing and engineering code-based cryptographic algorithms using these other kinds of codes efficiently is a prospective research direction to be explored’.

The above line has been included in line number 912.

Detailing further more on these research directions would not be possible in this paper because

  1. It could shift the focus of the paper which would be detrimental for the paper

These are research directions to be explored and hence only the possible details necessary to emphasize and establish that these are in fact highly prospective research directions have been given. Further details to these can only be provided after sufficient research has been carried out on each and not at the current stage itself.

This manuscript is a resubmission of an earlier submission. The following is a list of the peer review reports and author responses from that submission.

Round 1

Reviewer 1 Report

In this paper, the authors perform a survey on post-quantum cryptography, with a particular focus on code-based cryptography. The paper does not contain much novelty (it is a survey paper), but incorporates a fair description of the main papers and schemes which have contributed to bring code-based cryptography to the state it has nowadays.

Yet, I believe some important works and concepts are missing. Indeed, it is well known that the most relevant attack against code-based cryptosystems is ISD (Information Set Decoding), which have been introduced by Prange in 1962 (see ref. [A-D] for some of the most relevant algorithms, and [E] for a review of such algorithms for codes in the binary finite field). I think any survey on code-based cryptography must contain at least a subsection about these algorithms.

In the same way, I think some relevant schemes, which are also currently appearing among the candidates admitted to the third round of NIST PQC standardization process, are not properly mentioned. In particular, I'm referring to BIKE and HQC, which are based on random or pseudo-random QC codes. For these type of codes, ISD can be polynomially sped-up thanks to the QC structure (see ref. [F-H]), but this not dramatic. Indeed, it is well known that QC-MDPC codes can achieve very compact keys and are secure enough, speaking from a decoding point of view.

Yet, the most important issue with this kind of codes comes from so-called reaction attacks, which are based on events of decryption failures (a non null decoding failure probability is intrinsic to MDPC decoding). To counter this kind of attacks, a negligible failure rate (in the order of 2^-lambda, where lambda is the desired security level) is required, and this is why several recent works have exactly focused on devising decoders and parameters which achieve such a low value.

I think this also represent a major research direction in code-based cryptography, since devising such a reliable analysis for an MDPC decoder would represent a breakthrough in the area, allowing for secure (IND-CCA2 secure) and efficient schemes with very compact keys. This is also confirmed by the appearance of several works studying this exact problem.

I would briefly mention this aspect, speaking about reaction attacks (refs. [I-K]) and works studying the DFR of MDPC codes (all the other suggested references).

The section about digital signatures does not contain mentions on two important signature schemes (WAVE and Durandal), and may be expanded. Yet, I think there is no space for adding precise details about this topic, so I think the authors should declare (in some place) that they focus on encryption schemes.

Finally, I have spotted several typos; for this reason, I suggest the authors to carefully revise the manuscript (perhaps, with the help of a native english speaker), in order to improve its editorial quality.

##########################

SUGGESTED REFERENCES

[A] Prange, E. The use of information sets in decoding cyclic codes. IRE Trans. Inf. Theory 1962, 8, 5–9

[B] Stern, J. A Method for Finding Codewords of Small Weight. In Proceedings of the Coding Theory and Applications, 3rd International Colloquium, Toulon, France, 2–4 November 1988; pp. 106–113.

[C] Leon, J.S. A probabilistic algorithm for computing minimum weights of large error-correcting codes. IEEE Trans. Inf. Theory 1988, 34, 1354–1359

[D] Becker, A.; Joux, A.; May, A.; Meurer, A. Decoding Random Binary Linear Codes in 2 n/20: How 1 + 1 = 0 Improves Information Set Decoding. In Proceedings of the Advances in Cryptology—EUROCRYPT 2012—31st Annual International Conference on the Theory and Applications of Cryptographic Techniques, Cambridge, UK, 15–19 April 2012; pp. 520–536

[E] Baldi, Marco, et al. "A finite regime analysis of information set decoding algorithms." Algorithms 12.10 (2019): 209.

[F] Apon, Daniel, et al. "Cryptanalysis of Ledacrypt." Annual International Cryptology Conference. Springer, Cham, 2020.

[G] Sendrier, N.: Decoding one out of many. In: Yang, B.-Y. (ed.) PQCrypto 2011. LNCS, vol. 7071, pp. 51–67. Springer, Heidelberg (2011)

[H] Löndahl, C., et al.: Squaring attacks on McEliece public-key cryptosystems using quasi-cyclic codes of even dimension. Des. Codes Cryptogr. 80(2), 359–377 (2016)

[I] Guo, Q., Johansson, T., Stankovski, P.: A key recovery attack on MDPC with CCA security using decoding errors. In: Cheon, J.H., Takagi, T. (eds.) ASIACRYPT 2016, LNCS, vol. 10031, pp. 789–815. Springer Berlin Heidelberg
(2016) 

[J] Santini, Paolo, et al. "Analysis of reaction and timing attacks against cryptosystems based on sparse parity-check codes." Code-Based Cryptography Workshop. Springer, Cham, 2019.

[K] Eaton, E., Lequesne, M., Parent, A., Sendrier, N.: QC-MDPC: A timing attack and a CCA2 KEM. In: Lange, T.,
Steinwandt, R. (eds.) PQCrypto. pp. 47–76. Springer International Publishing, Fort Lauderdale, FL, USA (Apr 2018) 

[L] Baldi, Marco, et al. "A Failure Rate Model of Bit-flipping Decoders for QC-LDPC and QC-MDPC Code-based Cryptosystems." SECRYPT 2020-17th International Conference on Security and Cryptography. ScitePress, 2020.

[M] Tillich, J. (2018). The decoding failure probability of MDPC codes. In ISIT 2018, Vail, CO, USA.

[N] Santini, P., Battaglioni, M., Baldi, M., and Chiaraluce, F. (2019). Hard-decision iterative decoding of LDPC codes with bounded error rate. In Proc. IEEE Conf. on Communications (ICC 2019), Shanghai, China.

[O] Santini, Paolo, et al. "Analysis of the error correction capability of LDPC and MDPC codes under parallel bit-flipping decoding and application to cryptography." IEEE Transactions on Communications 68.8 (2020): 4648-4660.

[P] Sendrier, N. and Vasseur, V. (2019a). About low DFR for QC-MDPC decoding. Cryptology ePrint Archive, Report 2019/1434. https://eprint.iacr.org/2019/1434.

[Q] Sendrier, N. and Vasseur, V. (2019b). On the decoding failure rate of QC-MDPC bit-flipping decoders. In PQCrypto 2019, volume 11505 of LNCS, pages 404–416. Springer, Cham.

Reviewer 2 Report

This paper claims to provide a review of the post-quantum cryptography scenario, with an emphasis on code-based cryptography. In truth, it is very far from that. After reading through, it looks like the authors are trying to put together a survey based on other surveys, at best, and this ends up in a very superficial result.

Much of the material is presented vaguely, with very little insight into the area, and some very arbitrary choices, with important notions missing, or viceversa, needless details included. As is, I do not see how such a paper would be of interest. I recommend the authors to start over with an extensive literature review (reading research papers rather than surveys), and present the field of code-based cryptography with academic rigor and some meaningful insights. Some examples:

  • The entirety of section 4.1, presented as a list and without insight, is pointless
  • 4.2 is not relevant at all to such a survey paper
  • Several works on code-based signatures are omitted, and the entire scenario presented in few lines in section 4.5.2 is incomplete, misleading and useless, in the present form

Several other details need fixing, for example, Niederreiter is misspelled throughout the paper (Nierereitter), the notation is often pretty bad (especially for finite fields), references are missing etc.

Reviewer 3 Report

First of all, the reviewer did not finish reading all of the manuscript.  However, as it will take too long time for the reviewer to finish reading the whole and the reviewer has already found several points to be commented, the reviewer would like to submit the first review report at this moment (rather than letting the authors wait for further long time).

The survey paper under review is entitled "Code-based Post-Quantum Cryptography" but Section 2 of this paper is devoted to general quantum computation and Section 3 is devoted to an overview of post-quantum cryptography, not just code-based ones.  The part specific to code-based cryptography will start from Section 4.  And the reviewer has read this paper until Section 3.2 (including Table 2 on page 7), but the reviewer has found several comments as mentioned above.  To be honest, these points to be commented made the reviewer feeling that the quality of this survey paper is not so high (and sometimes, feeling curious about whether the authors are indeed experts of this area enough for writing a survey article in a peer-reviewed journal).

Major Comments:

  • Abstract, line 7: "... could be solved in deterministic time" would be "... in polynomial time" (at least, the present expression would not make sense for the reviewer).
  • p.1, Sect.1, line 2 (Line 16 in total): For the reference [2], it seems not reasonable for the reviewer to refer to this article.  The present sentence is mentioning about factoring-based or DL-based cryptography, but the reference [2] looks a paper on code-based cryptography.  (As this is a survey paper, the references should be chosen more carefully than usual research papers.)
  • p.1, 8th to 5th last lines (Lines 29-32 in total): The authors wrote that quantum algorithms would not much affect the security of symmetric key cryptography.  However, in fact a family of research results started from [Kuwakado and Morii, ISIT 2010] are suggesting potential risk for symmetric cryptography caused by quantum attacks.  So the present expression is misleading and therefore should be revised. 
  • p.3, 1st line (Line 88 in total): O(\sqrt{N}) is a query complexity of Grover's algorithm, not time complexity.
  • p.3, last line before Sect.3.1 (Line 107 in total): The authors refer to a page in Wikipedia (reference [19]).  The reviewer thinks that such a reference to an unstable resource (like a web page) is allowed only when the reference is necessary and cannot be replaced by any other stable resource (like a book or journal/conference paper).  There seems no essential reason that Wikipedia page is referred to in this survey article.
  • p.3, paragraph "Hash-based cryptography", 3rd line (Line 115 in total): For the expression "(as one-way function)", the security of hash-based schemes is based on not just one-wayness of the hash function, but also collision-resistant property and hardness of second pre-image attacks.  Therefore the current expression is too misleading.
  • p.3, paragraph "Code-based cryptography", 1st line (Line 124 in total): Again, for reference [23], there seems no essential reason to just refer to a PDF file on a web page; there should be many good books or papers on code-based cryptography that are worthy to be referred to.
  • p.3, last two lines of paragraph "Code-based cryptography" (Lines 129-130 in total) (and also some other places in the paper): The authors sometimes refer to finalists/alternate candidates of the third round of NIST PQC standardization as "outcome" (or "result", etc.) of the third round.  However, the third round is now ongoing and nothing has come out from the third round.  Therefore, such expressions should be revised.
  • p.3, paragraph "Multivariate cryptography", 1st line (Line 132 in total): The term "multivariate-based cryptography" is not commonly used.  Just writing "multivariate cryptography" is better.
  • p.3, last line (Line 136 in total): When mentioning about the UOV scheme, a reference to the original paper of UOV should be written (as this is a SURVEY article).  Similar comments are applied to many other places in the paper, such as "Rainbow", "TTS", and "MPKC" in the next line. 
  • p.5, Table 1: There are several mistakes and unclear points in the table.  For example, in the first row:
    • 1st cell: It is not clear what "S. No." stands for.
    • 5th cell: "Excha- nge" should be a typo.
    • 6th cell: Isn't it Public Key "Size"?
    • 7th cell: Isn't it Private Key "Size"?
    • 8th cell: "Signa- ture" should be a typo.
    • 11th cell: What does "Li bo qs" mean?
  • Also, in Table 1, 4th and 5th rows, for "Weaknesses": Even if "Only signature scheme is available" is true, this is not a reason to write "not a complete cryptosystem".  (What does "not a complete cryptosystem" mean?)
  • p.7, Table 2, No.3, for "Algorithms used": SIKE is "supersingular isogeny key exchange", not just "supersingular isogeny".

Minor Comments:

  • p.2, 2nd last line (Line 86 in total): "%including" should be a typo.
  • p.4, 4th line (Line 140 in total): "one of the finalist" -> "one of the finalists"
  • p.4, 5th line (Line 141 in total):  "alternate finalist" should formally be "alternate candidate".  Also, "scheme" -> "schemes"
  • p.4, paragraph "Lattice-based cryptography", 5th last line (Line 151 in total): In "one of the successful scheme", "scheme" -> "schemes"
  • p.4, Sect.3.2, 2nd line (Line 176 in total): The last "available" should be followed by something like a period, colon, etc.
  • p.7, Table 2: An overflow is occurring for the cell containing "Switzerland".

Reviewer 4 Report

This paper provides a review of the various post-quantum cryptography, and give an overview for code-based cryptography. This paper also provides some research directions that are not yet explored in code-based cryptography. This paper also summarizes the recent candidates of post-quantum cryptography (PQC) available in each of the PQC techniques, and compares the PQC solutions. This papers details the industry surveys of PQC initiatives and the standardization efforts in PQC.

Overall, I think the content of this work does not fit well with the title of this paper. The title of this paper is “Code-based Post-Quantum Cryptography”. However, I notice that 9 out of 24 pages (about 40%) of the content is about PQC, which includes the industry survey and standardization efforts of PQC (which includes lattice-based, code-based, isogeny-based, and etc). I suggest the authors to make a decision of their choice of content: either to talk mainly on all the PQC schemes, or just focus on code-based cryptography. Should the authors decide to focus on code-based cryptography, then I suggest they cut down the parts on Section 2 and Section 3, and try to expand Section 4 and 5 (more on code-based cryptography).

Also, I find that Pg. 17, Line 571-586 are somewhat awkward to me when they appear here. Shouldn’t these two paragraphs be included in Section 1 under the authors’ contribution?

Next, a major comment on “code-based cryptography”. I understand that this paper is supposed to be a survey paper, but I find that the descriptions of codes (and related concepts) can be improved. Moreover, I do not think that the authors displayed a comprehensive understanding of code-based cryptography, but rather they put efforts on the encoding perspectives and relationships between codes.

In particular, the authors have mentioned quite a number of error-correcting codes in 4.1, but some of the codes are not used in cryptography. Furthermore, in Section 4.5, the authors claimed that “code-based cryptosystems provide for code-based cryptography and code-based signature schemes”. I would like to point out that the system constructed in code-based cryptography are called code-based cryptosystems. And there are two (or three) main types of cryptosystems: 1. Public-key Encryption/Key Exchange/Key Encapsulation; 2. Digital signature schemes. If the above sentence by the authors is not a typo, then I suggest the authors to improve their understanding on code-based cryptography.

Furthermore, in Pg. 11 Line 344, the authors claimed that “In Code-based Cryptography, only binary codes are considered i.e. codes over F2.” This statement itself is not true, as there are some other finite fields being considered in public-key encryption schemes based on codes. Even if this statement is true, the authors are self-contradictory, as they mentioned the usage of Gabidulin codes, GRS codes and etc in Table 5. In addition to that, there are actually some other codes being used in Niederreiter system, rather than only GRS code being used.

As the authors claimed that their comparisons are novel, I suggest the authors to update some of their knowledge on the public-key encryption schemes and signature schemes. I will just give some examples: for PKE, there are some latest constructions (besides those proposed to NIST PQC Standardization) such as subcode subfield of Reed-Solomon codes, IKKR cryptosystems, and etc. For signature schemes, the authors can consider WAVE signature scheme.

As I mentioned before, the authors can expand the content of code-based cryptography, in particular to discuss more on the attacks on code-based cryptosystems (can even make it to be one section).

Besides that, I think Section 4.3 should be renamed to “Linear codes endowed with different metrics”. In fact, besides endowing linear codes with classical Hamming metric, there are quite a number of significant and important works on code endowed with rank metric codes. As such, the Gabidulin codes in Table 5 should be discussed separately. Also, it is worth mentioning that Lee metric can be used for code-based cryptography as well.

Finally, my next major comment for this paper is: I agree with the authors that what they argued in Section 5 are indeed future research directions for code-based cryptography. However, I do not think that these are important at the moment, because for code-based cryptography, we do not even have a secure encryption scheme or signature scheme with compact key size, lest to talk about their applications in digital systems, encoding and etc. Therefore, I think the immediate challenge is to design secure code-based cryptosystems with compact key sizes.

There are quite a number of typos and misspell in the paper. Please check the grammar as well.